# Multi-Perspective Views and Hesitancy toward COVID-19 Vaccines: A Mixed Method Study

**DOI:** 10.3390/vaccines11040801

**Published:** 2023-04-05

**Authors:** Serine Sahakyan, Natella Gharibyan, Lusine Aslanyan, Varduhi Hayrumyan, Arusyak Harutyunyan, Lorky Libaridian, Zaruhi Grigoryan

**Affiliations:** 1Turpanjian College of Health Sciences, American University of Armenia, Yerevan 0019, Armenia; 2Cambridge Health Alliance, Harvard Medical School, Cambridge, MA 02139, USA

**Keywords:** vaccine, COVID-19, healthcare provider

## Abstract

The worldwide uptake of COVID-19 vaccines was suboptimal throughout the pandemic; vaccine hesitancy played a principle role in low vaccine acceptance both globally and in Armenia. In order to understand the factors behind the slow vaccine uptake in Armenia, we aimed to explore the prevailing perceptions and experiences of healthcare providers and the general public related to COVID-19 vaccines. The study applied a convergent parallel mixed-methods study design (QUAL-quant) through in-depth interviews (IDI) and a telephone survey. We completed 34 IDIs with different physician and beneficiary groups and a telephone survey with 355 primary healthcare (PHC) providers. The IDIs found that physicians held variable views on the need for COVID-19 vaccination which, combined with mixed messaging in the media landscape, fueled the public’s vaccine hesitancy. The survey results were mostly consistent with the qualitative findings as 54% of physicians hypothesized that COVID-19 vaccines were rushed without appropriate testing and 42% were concerned about the safety of those vaccines. Strategies to improve vaccination rates must target the main drivers of hesitancy, such as physicians’ poor knowledge of specific vaccines and spiraling misconceptions about them. Meanwhile, timely educational campaigns with targeted messaging for the general public should address misinformation, promote vaccine acceptance, and empower their capacity to make decisions about their health.

## 1. Introduction

Since March 2020, the world has been battling the pandemic of Coronavirus Disease 19 (COVID-19) [1]. The mobilization of worldwide resources resulted in the development of safe and effective vaccines within an unprecedentedly short period of time [2,3]. Vaccines carry an enormous public health value in modern healthcare, cost-effectively advancing global welfare [4,5,6], and are considered to be one of 10 great public health achievements of the 20th and 21st centuries [7,8]. The approved vaccines against COVID-19 are proven to be highly effective and safe in preventing moderate to severe forms of the disease [9,10,11].

Despite strong evidence demonstrating the safety and effectiveness of vaccines against COVID-19 and its severe outcomes, the worldwide uptake of COVID-19 vaccines remains insufficient, with Armenia being no exception. Despite the rapid development of the vaccines, and while the pandemic progresses global COVID-19, vaccine coverage is far from satisfactory. As of 30 January 2022, due to limited access to vaccines, the poorest countries of the world, which comprise one-fifth of the world’s population, account for only 6.4% of global vaccinations [12]. At the same time, countries with sufficient amounts of vaccine still struggle to establish levels sufficient for herd immunity. The United States has only a quarter of its population fully vaccinated [13], and nearly 30% of the European population has not yet received even the primary vaccine course [14].

Barriers to vaccine uptake are multi-dimensional, including structural (cost, convenience, supply chain issues) and attitudinal (perceived risk, trust, misinformation, misconception) factors [15,16]. Vaccine hesitancy, among other factors, plays a principle role in low vaccine acceptance [17,18]. The World Health Organization (WHO) defines vaccine hesitancy as “delay in acceptance or refusal of vaccines despite availability of vaccine services” [17]. The most common determinants of vaccine hesitancy include context-specific safety-related concerns, mistrust, and low health literacy [19,20,21,22]. Additionally, healthcare providers, particularly at the primary care level, remain the most influential figures in public endorsement of vaccination, while low confidence and lack of training in vaccination in this group decreases the potential to overcome public vaccine hesitancy [22,23,24].

Since the pandemic started, the Republic of Armenia (Armenia) has experienced five COVID-19 waves [25,26]. As of December 2022, in total, more than 445,000 confirmed COVID-19 cases and almost 8700 deaths have been recorded in the country [25].

Armenia, similar to many countries worldwide, has been struggling to reach sufficient vaccine coverage against COVID-19. Vaccination against COVID-19 became freely available in April 2021 in Armenia, initially targeting certain high-risk groups [27]. As of May 2021, COVID-19 vaccines were available to the entire adult population including citizens, residents, and travelers [28]. At the time of the study (July–September 2021), vaccine coverage was still lagging with only around 4% of the population having received at least one dose as of mid-July [29]. During the same time period, countries with similar socio-demographic profiles (e.g., Georgia, Estonia) had comparatively better progress in vaccine coverage [30].

The Ministry of Health (MoH) established a working group to coordinate a public awareness campaign when vaccines became available in Armenia. Comprised of over ten national and international organizations, it focused on developing risk communication tools, and social media campaigns to promote vaccination. The campaign messaging was delivered through various channels including ads on TV and social media. The government also initiated a series of healthcare provider training focusing on the importance of vaccination, brand-specific aspects of vaccine administration, and possible side effects. The trainings particularly targeted physicians and nurses from the primary health care level. In October 2021, a governmental decree required mandatory testing twice a week or full vaccination for all employees and students. This strategy kicked the vaccination coverage to 45.9% as of December 2022. Despite the availability of several types of COVID-19 vaccines at free of cost, the coverage did not reach the recommended threshold of 90%. Furthermore, the population groups with the lowest vaccine coverage remained older adults, and people with chronic conditions [31].

To better understand the challenges and factors behind low COVID-19 vaccination coverage in Armenia, we aimed to explore the prevailing perceptions and experiences of healthcare providers and the general public related to COVID-19 vaccines. The specific objectives were to qualitatively investigate the attitudes and concerns of the general public and of physicians related to COVID-19 vaccines, to quantitatively assess the vaccination status of primary healthcare (PHC) providers and their attitudes toward vaccination, and to assess their readiness to advise patients to get vaccinated against COVID-19.

## 2. Materials and Methods

### 2.1. Study Design

The study applied a convergent parallel mixed-methods study design with a dominant qualitative phase (QUAL-quant). The qualitative component explored the perceptions and concerns of the general public and of physicians related to COVID-19 vaccines using semi-structured in-depth and paired interviews (IDIs). At the same time, a cross-sectional phone survey was conducted with PHC providers, and the data on COVID-19 vaccine intake and readiness were obtained from the surveys to triangulate the qualitative findings. The use of open-ended questions and multiple probes, in contrast to structured surveys, provided an opportunity to thoroughly investigate the underlying dimensions of the main topics and to obtain rich data about the participants’ overall perceptions of COVID-19 vaccination [32].

### 2.2. Study Setting and Participants

The qualitative component included participants from all over Armenia, including different marzes (regions) and Yerevan, the capital. To obtain a broad spectrum of perspectives, different groups of participants were selected including physicians and the general public. The group of physicians included general practitioners, family doctors, and different specialists providing essential health services (e.g., pediatricians, gynecologists, endocrinologists, infection prevention, and control specialists) at PHC facilities (hereafter physicians). The group of general public included residents of Armenia and beneficiaries of the PHC essential health services (e.g., patients with chronic heart disease and diabetes, and mothers of children at the child immunization stage) (hereafter general public).

As for the quantitative component (phone survey), the target population was general practitioners and family doctors (hereafter PHC providers) from PHC facilities. The inclusion criterion was having been involved in the outpatient management of COVID-19 patients in Armenia for at least 1 month.

### 2.3. Study Instruments

For the qualitative interviews, a semi-structured interview guide was developed to more deeply explore participants’ multiple perspectives and attitudes toward COVID-19 vaccination, their perceptions of the importance of getting vaccinated, and their thoughts on COVID-19 vaccine effectiveness and safety. The guide was comprised of five open-ended questions with corresponding probes (Table 1). It was developed in English, translated into Armenian, and pre-tested. Based on the experience of the initial IDIs, the interview guide was revised to improve the flow and formulation of the open-ended questions. The interviewers also collected the participants’ demographic information (e.g., age, gender, region).

The survey instrument contained several domains including sociodemographic characteristics of participants (age, gender, years practicing as a physician, region); COVID-19 vaccination status and number of received doses; attitude toward vaccines (11 items with answer options “Strongly agree”, “Agree”, Disagree”, and “Strongly disagree”); COVID-19 risk perception (whether they think that they are at risk of getting COVID-19 in the next 1 year, the reasons why they do not think they will get COVID-19 or how severe they think their COVID-19 infection would be); readiness to advise their patients to get vaccinated with answer options “Yes”, “No”, “Not sure”. The survey instrument was set up in the Alchemer online tool which allowed simultaneous data entry [33].

### 2.4. Participants’ Sampling and Data Collection

For the qualitative interviews, the research team used a theoretical sampling approach and purposefully recruited participants to further explore the concepts that emerged from the preliminary analysis of each completed interview [34]. We approached the physicians through the administration of the PHC facilities, whereas the participants for the general public group were identified through the personal networks of research team members.

The data were collected during May–September 2021. Overall, 32 IDIs were conducted, of which two were paired (dyadic) [35]. We interviewed 18 physicians and 16 participants from the general public. The mean duration of an IDI was 42 min, while the paired interviews lasted for more than an hour.

Four researchers conducted most of the IDIs remotely, utilizing different virtual platforms. All of them were video-assisted to foster rapport building. Following the participants’ preferences, 10 interviews were conducted face-to-face. All interviews were audio recorded after receiving participant consent. If participants refused to be audio-recorded, then detailed notes were taken. The interviews were stopped after reaching data saturation.

For the survey, the sample size was calculated to be 384 using the one-sample proportion formula [36]. The research team used the simple random sampling technique to recruit participants from the complete list of all PHC providers developed by the National Institute of Health after Academician S. Avdalbekyan (NIH).

To reach the desired sample size, the research team contacted 519 potential participants. Of those, 24.9% (*n* = 129) refused to participate (the main reason for refusals was the lack of time and busy schedules), 5.0% (*n* = 27) were ineligible, and 1.5% (*n* = 8) partially completed the survey. As a result, the study team was able to complete 355 interviews with a 72% response rate.

### 2.5. Data Analysis and Study Rigor

The qualitative interviews were transcribed verbatim along with data collection in the native language. Iterative thematic analysis was performed [37], coding the textual data and categorizing based on inductively emerging categories. Three team members performed the coding of the transcripts. During the weekly team meetings, individual cases, codes, and identified patterns were discussed and agreed upon.

The survey data were extracted from Alchemer and were cleaned and analyzed using Stata 13 software (StataCorp. 2013. Stata Statistical Software: Release 13. College Station, TX, USA: StataCorp LP). In the paper, data were presented using counts and percentages.

All of the interviewers for IDIs were public health specialists. Additionally, the researchers who interviewed the physicians were specialized in internal medicine and pediatrics, and the interviewer of the participants from the general public was a social worker. The findings of the thematic analysis were enriched by the descriptive quantitative data resulting from the phone survey. Collecting triangulated data in terms of the data source, data collection methods, and investigators, the study aimed to strengthen its quality by enriching and validating the results.

## 3. Results

### 3.1. Socio-Demographic Characteristics of the Study Participants

Of the 18 physicians who participated in the qualitative component, 17 were women. Seven of them were working at PHC facilities in the capital; the remainder were from six different marzes (Syuniq, Tavush, Aragatsotn, Armavir, Ararat, and Lori) of RA. The majority of the IDI participants in the general public group were also women (12 out of 16), with a mean age of 49.1 years, ranging from 23 to 75 years.

The vast majority of the PHC providers (92%) who participated in the survey were female, with a mean age of about 56 years. The number of urban participants was slightly higher (58%) compared to rural participants. The average duration of being involved in treatment of the COVID-19 patients among PHC providers was 16 months. The reported length of experience as a PHC provider was approximately 25 years on average (see Table 2).

### 3.2. Themes

Three main themes were identified and labeled as follows: (1) Variable views among physicians on the need for COVID-19 vaccination; (2) mixed messaging fueling hesitancy for physicians and the general public; (3) the public’s concerns regarding vaccination safety. The themes are presented in combination with the relevant quantitative findings.

#### 3.2.1. Theme 1: Variable Views among Physicians on the Need for COVID-19 Vaccination

Though most of the physicians generally held positive views on COVID-19 vaccines and acknowledged their effectiveness, for some of them the vaccination was not their first choice for prevention of COVID-19 infection. Those who believed that vaccines are a necessity to overcome the pandemic justified it as “


*“…the first preventive measure for all types of infections.”*
Physician 12


*“…vaccines are necessary because at this point we don’t know how to get out of the epidemic.”*
Physician 2

Some of the physicians reported being vaccinated, while others were planning to get one in the future.


*“I’m already vaccinated, the first [dose]. Soon I will get the second one.”*
Physician 3


*“I will get the vaccine for sure․”*
Physician 2

Those physicians who expressed some reservations about vaccination and its effectiveness cited other measures to fight COVID-19, including preferring natural infection over vaccination. For instance, a few participants raised doubts about the usefulness of vaccination, preferring natural infection.


*“I think if 60–70% [of society] has immunity [having COVID-19 antibody as a result of contracting the disease], vaccinations will not be as effective, they can even have an adverse effect.”*
Physician 12


*“In any case, it is wrong to interfere too much with the immune system. Human immunity should be allowed to function for itself.”*
Physician 1


*“We have already been infected, what is the need for vaccination?”*
Participant from general public 5


*“I know someone who was vaccinated and is currently hospitalized. It has been a month and I am not sure what will happen. That person also had COVID, took it very lightly, but now is in a very serious condition because of the vaccine.”*
Participant from general public 7

Another participant stressed the predominant role of social distancing and hygiene measures for effective infection control, rather than vaccination.


*“It seems to me that the most effective/important measure is not the vaccine but reducing the spread of infection… through social distancing and wearing of masks.”*
Physician 4

The survey among PHC providers was mostly consistent with the qualitative findings. Almost all of them (96%) strongly agreed or agreed that COVID-19 vaccines can significantly reduce the spread of the pandemic. Furthermore, for the vast majority (95%) of those surveyed, COVID-19 vaccines were the best way to prevent disease complications. However, only about 63% of PHC providers in the survey reported being fully vaccinated against COVID-19, only 73% of them preferred vaccination over natural immunity, and 7% were still against vaccines in general. When asked about their prior exposure to COVID-19, 4.4% of the survey participants indicated having had the disease and being immune to it and 29.5% stated that they have already recovered from the disease and will not get re-infected (see Table 3).

#### 3.2.2. Theme 2: Mixed Messaging Fueling Hesitancy

Physicians and participants from the general public reported different levels of vaccine refusal and different degrees of counseling provided by healthcare providers. Participants’ non-readiness to get vaccinated ranged from hesitancy to clear rejection.

The concerns and hesitancy voiced by the great majority of participants from the general public and from physicians reflected the influence of controversial news and misinformation.


*“It is just a little scary, every country says different things, for instance, that a certain vaccine causes thrombosis. All this makes me afraid...“*
Participant from general public 8


*“According to some statistics, after listening to some doctors, reading and listening to some interviews, I realized that during some autoimmune diseases, you can’t be vaccinated. Now I don’t know whether it is right to get vaccinated or not.“*
Participant from general public 10

Moreover, the participants’ responses indicated that hesitancy to get vaccinated was further fueled by misleading advice and by the lack of clear guidance from physicians: A significant portion of this revolved around questions regarding the safety of the vaccines. This finding was widely triangulated across participants’ experiences.


*“I say to my doctor “I want to get vaccinated, which one [would you advise]”? She says “I find it difficult because it hasn’t been long, five years have not passed yet.”“*
Participant from general public 4


*“What consequences vaccines will have and what positive or negative effects they will havewill be known several years from now.“*
Physician 5


*“A patient comes and says “Are you sure nothing will happen to me?” You can’t say “Yes, nothing will happen.”“*
Physician 8

For instance, one participant shared her experience when her and her husband’s decision to be vaccinated was perceived as a risky step by many healthcare providers in their PHC facility.


*“Everyone was surprised [physicians]… They found it very strange that we decided to get vaccinated and take the risk.“*
Participant from general public 5

While describing the PHC providers’ role in the decision to get vaccinated, some participants from the general public named them as playing an important role. One participant, with an underlying health condition, reflected on being encouraged by her PHC provider to get the COVID-19 vaccination.


*“My doctor—general practitioner—said: “it is better for you to get vaccinated.“*
Participant from general public 8

Another participant added, *“The doctor said that it is the right thing [to do] to get vaccinated… especially given that I am in a risk group [for severe COVID-19 disease and outcomes]. He explained it to me in a clear way.”*Participant from general public 9

The PHC providers and specialists also acknowledged their role in influencing the decision to get vaccinated:


*“They [patients] come to us and ask about vaccinations. They say “First, we would like to talk to you and then decide whether to get vaccinated or not, to understand which vaccine to choose, and so on”.“*
Physician 9

Similarly, the vast majority of the surveyed PHC providers (98%) stated that they would advise their patients to get vaccinated against COVID-19 (see Table 3). Moreover, the majority of them thought it should be mandatory for everyone who is eligible (77%) and for all healthcare providers (80%).

#### 3.2.3. Theme 3: The Public’s Concerns Regarding Vaccination Safety

Some participants from the general public strongly perceived vaccination as a “risky” step. They commonly expressed fear and concerns over possible side effects and unforeseen consequences of the vaccination.


*“I will always feel like I am going to develop a thrombosis, or I will have a heart problem [because of the vaccination]. I’m not someone who is easily scared but I am really scared of this.”*
Participant from general public 1


*“Oh no, I’m afraid, I have a few problems, I have some allergy related issues. I’m afraid of getting vaccinated.”*
Participant from general public 2


*“Maybe in twenty years [after the vaccination] I will develop a disease and no one will understand where it came from.”*
Participant from general public 6

Those who reflected on their concerns regarding the risks of COVID-19 vaccines tended to draw on others or their personal past experience while framing their anxiety. Participants noted knowing or hearing about vaccinated people with adverse outcomes.


*“One of our acquaintances said that someone got vaccinated. He was perfectly healthy and died a week later. He had no problems, no heart problems, no diabetes, nothing. I don’t know what to say.”*
Participant from general public 5


*“I know a person who got vaccinated and now is in a very severe [medical] condition. He has been in the hospital for one month now and I don’t know what will happen.”*
Participant from general public 8

On the other hand, some PHC providers, speaking about public anxiety toward the COVID-19 vaccination, considered it expected as it had happened with previous vaccines, as well.


*“It was the same with Gardasil in the beginning. Now parents responsibly bring [their children to get the vaccine]. However, in the beginning, I could have 25 people in 3 villages, out of which 1 would get vaccinated or not. Now everyone gets vaccinated, and only one of them might refuse …”*
Physician 3

The survey findings among PHC providers were largely in line with the qualitative findings. Specifically, more than half of the survey participants (54%) agreed or strongly agreed that COVID-19 vaccines were being rushed without appropriate testing (see Table 3) and 42% of them were concerned about the safety of a vaccine developed emergently during the pandemic. In fact, most PHC providers believed in the effectiveness of the COVID-19 vaccine: More than half of them (58%) perceived themselves not-at-risk for contracting COVID-19 during the subsequent year due to having been vaccinated (57%). Additionally, the vast majority of survey participants still reported that they trust science to develop safe (97%) and effective (98%) new vaccines.

## 4. Discussion

The findings of the current study revealed a spectrum of perceptions and concerns related to COVID-19 vaccines among physicians and the general public. In general, three large themes emerged from the data: Variable views on COVID-19 vaccination among physicians; the impact of mixed messaging on vaccine acceptance; and participants’ fears about the “new” vaccine.

In our study, physicians reported favorable opinions toward COVID-19 vaccination, in the majority of cases expressing trust in the benefits of the vaccines. However, when it came to their personal vaccination status, some of them reported being vaccinated, while others shared their plans to become vaccinated in the future. The survey results showed that only 65% of the physicians were vaccinated, even though the vaccines were widely available in the country [38]. This discrepancy between the expressed positive attitude and the real personal practice may indicate a hesitancy among physicians and/or a social desirability bias that caused them to express expected ideas. A study conducted in Denmark revealed that the average Measles, Mumps, and Rubella (MMR) vaccination rate among doctors with a positive attitude toward vaccination was 85%, compared to 69% in practices with more reserved attitudes [39].

Interestingly, physicians expressed extremely varying attitudes toward vaccines. Their arguments in favor of alternative measures—such as natural immunity, social distancing, and hygiene measures—were concerning. The majority of surveyed PHC providers mentioned trusting science to develop vaccines and advising their patients to get the COVID-19 vaccines. Yet, surprisingly nearly half of them agreed with the contradictory statement that the development of COVID-19 vaccines was rushed without appropriate testing. The literature shows varying patterns of perception regarding COVID-19 vaccination among healthcare providers, ranging from full acceptance to total refusal [40,41,42,43]. The inconsistencies and controversies that arose in our study in the attitudes toward COVID-19 vaccines are likely multifactorial. For example, the background, education, and professional practice of healthcare providers likely predisposed them to trust science and to consider vaccines an effective measure against infectious diseases. However, knowledge gaps regarding COVID-19 vaccine development on the one hand, and consistent and targeted misinformation from the media against COVID-19 vaccines on the other, might have triggered doubts and hesitancy. The idea that the COVID-19 vaccines had not yet undergone proper testing was amongst the most frequently cited arguments against COVID-19. In our findings, the influence of misinformation was also recognized by the participants.

Participants cited others or personal past experience as reasons for their hesitancy. These experience-based stories were often very vague and based on events surrounding one or a small number of individuals [44]. This type of reliance on anecdotal information rather than data is common in Armenia, and could make it easier for fear and misinformation to spread [45].

Additionally, our findings highlighted the important role that physicians play in patients’ decisions. The experiences of all groups of participants indicate that physicians are often approached for advice on vaccination. The essential role of the healthcare workers, and especially PHC providers in promoting vaccination, is indisputable and is widely acknowledged in the literature [22,23,24,46], though identifying what sources are the most trusted for any given population is key, given there is variability [47]. The inconsistency of physicians’ attitudes and practices as well as their role in relation to COVID-19 vaccination might have challenged Armenia’s progress toward greater COVID-19 vaccination acceptance.

In addition to listening to their physicians, the general public was listening to the media, including news and messaging from other countries. The variability in messages amongst all of these contributed to distrust and hesitancy regarding vaccination. Studies have found that inconsistent risk messages regarding COVID-19 vaccines may reduce vaccine uptake [48] for the general public, and that mixed messaging can have a negative impact on healthcare workers’ recommendation regarding vaccines [49].

Our study demonstrated that the general public’s rejection of COVID-19 vaccination was due to their fears and concerns over possible side effects and unforeseen consequences of the vaccination. In addition, misleading and contradictory information from different sources further triggered their fears and distrust. Physicians in our study acknowledged that historically vaccines have not been easily welcomed by the general population and considered these reactions as expected. Numerous studies have discussed that common fear and anxieties negatively affect vaccine acceptance throughout the world [15,16,19,21,22,50,51,52]. In the literature, the lack of knowledge, the spread of misinformation, and skepticism are found to be instrumental in contributing to vaccine hesitancy among the general public [4,15,18,19,21]. Kricorian et al. found that people who thought the COVID-19 vaccine was unsafe were less likely to get the shot, knew less about the virus, and were more likely to believe the myths about the vaccine [53]. In contrast, some studies indicate that individuals who understand that coronavirus is easily spread and deadly, have high health literacy, and have had positive experiences with vaccines in the past, are more likely to accept COVID-19 vaccines than others [54,55,56,57,58,59,60]. Therefore, targeting vaccine hesitancy should mainly include ensuring accurate information [54].

### Limitations

There were several limitations in this study. As the study participants chose whether or not to participate, there was a chance of self-selection bias. However, the use of multiple data sources has minimized this bias. Some study participants may also have provided more socially desirable answers; therefore, the true situation might be worse than described. Although the study team applied several measures (triangulation, member checking, collecting data in different geographical areas) to enhance the rigor of the study, researcher bias (related to correct interpretation of the findings) might still have influenced the results. In addition, the inclusion of only physicians, and not nurses, is another limitation of the study. This is especially true given the significant role that nurses play in patient communication and engagement, especially in rural Armenia.

Another limitation of the study could be the fact that the study team could not reach the predefined sample size due to many contact-related issues and a high refusal rate.

## 5. Conclusions

Strategies to improve vaccination rates in Armenia must target the main drivers of hesitancy: Lack of knowledge of the general public and healthcare providers, mixed messaging by providers and the healthcare system in general, and the specific fears and misinformation of the general population. The overall strategy is more likely to be successful if, while aligned in its content, it is designed with the needs of two different audiences in mind: The general population and healthcare providers. In light of the fundamental role that healthcare providers play in improving vaccination rates, and conversely, harm that under- and misinformed providers cause, proper communication with and training of healthcare providers is a core component of any strategy. More specifically, improving physicians’ knowledge about specific vaccines—including their effectiveness and safety, and knowledge and strategies regarding common misconceptions about vaccines—can increase their confidence in vaccines and their willingness to recommend them to others [24]. Furthermore, patients should not be considered as passive recipients of care, but have the capacity (health literate and informed) to make decisions regarding their health. Therefore, physician trainings should include communication skills and shared decision making training. Additionally, organized awareness raising campaigns that provide clear information about the safety and efficacy of vaccines and the technology used in their production should be implemented targeting the general public in order to build trust toward the vaccines [61]. Timely educational campaigns are crucial for diminishing vaccine hesitancy, increasing vaccine acceptance, and correcting misinformation about the vaccine.

Our study results are aligned with the international literature: A coordinated communication plan is fundamental for improved vaccine uptake [29]. While specifics related to a new vaccine or infectious agent may not be available until later into a pandemic, countries can and should have ready, well-designed, and easily implementable plans for developing and spreading consistent messaging to target populations, even before the next pandemic. This strategic plan should be part of a larger national pandemic plan for any country, and would be activated even before specifics are ready. This early mobilization will increase chances of higher vaccine uptake, which in turn, will reduce morbidity, mortality, the burden on the healthcare system, and contribute to the end of the next pandemic.

## Figures and Tables

**Table 1 vaccines-11-00801-t001:** Interview guide on attitude toward COVID-19 vaccination, readiness to get vaccinated, and vaccination status.

	Question
1	*What have you heard about the vaccine against COVID-19?*
2	*What are your thoughts on the effectiveness and safety of the vaccines?*
3	*What do you feel about the importance of COVID-19 vaccination?*
4	*Have you thought about getting vaccinated against COVID-19?*
5	*What do you think about the vaccination program adopted by the Armenian government?*

**Table 2 vaccines-11-00801-t002:** Socio-demographic characteristics of the PHC providers who participated in the survey.

Variables	Total *n* = 355
Gender, n (%)	
*Female*	325 (91.6)
Region, n (%)	
*Urban*	206 (58.0)
*Rural*	149 (42.0)
Age (years), m (SD)	56.3 (11.2)
Number of months involved in COVID-19 outpatient treatment, m (SD)	15.5 (2.3)
Number of years practicing to be a PHC provider, m (SD)	24.7 (13.5)

**Table 3 vaccines-11-00801-t003:** PHC providers’ perceived risk of getting COVID-19, vaccination status, attitude toward COVID-19 vaccination, and readiness to advise vaccination to their patients.

Variables	Total *n* = 355 n (%)
**Do you think you are at risk of getting COVID-19 in the next 1 year?**	
*Yes*	136 (42.2)
**Why do you think you are not at risk of getting COVID-19 in the next 1 year?**	
*I believe I already had the disease and I am immune to it (not diagnosed by a test)*	8 (4.4)
*I have already recovered and will not get re-infected (diagnosed by a PCR test)*	54 (29.5)
*I am vaccinated against COVID-19*	105 (57.4)
*I did not have clinical symptoms but I have antibodies against COVID-19*	5 (2.7)
*Other*	11 (6.0)
**What do you think, how severe will your COVID-19 infection be?**	
*I will get mild symptoms which will probably not require hospitalization*	53 (58.2)
*I will get moderate symptoms which will probably need hospitalization*	21 (23.1)
*I will get severe symptoms which will probably require admission to the intensive care unit*	17 (18.7)
**Have you received COVID-19 vaccine**	
*Yes*	232 (65.4)
**How many doses of COVID-19 vaccine have you received?**	
*1 dose*	86 (37.1)
*2 doses*	146 (62.9)
**In general, I am against vaccines.**	
*Strongly agree or Agree*	26 (7.4)
*Strongly disagree or Disagree*	326 (92.6)
**COVID-19 vaccines can significantly reduce the epidemic spread.**	
*Strongly agree or Agree*	335 (96.3)
*Strongly disagree or Disagree*	13 (3.7)
**COVID-19 vaccines are the best way to prevent disease complications (e.g., hospitalization, pneumonia)**	
*Strongly agree or Agree*	333 (94.9)
*Strongly disagree or Disagree*	18 (5.1)
**A COVID-19 vaccination should be mandatory for everyone who is able to receive it.**	
*Strongly agree or Agree*	269 (76.6)
*Strongly disagree or Disagree*	82 (23.4)
**A COVID-19 vaccination should be mandatory for all healthcare providers who are able to receive it.**	
*Strongly agree or Agree*	283 (80.2)
*Strongly disagree or Disagree*	70 (19.8)
**COVID-19 vaccines are being rushed without appropriate testing.**	
*Strongly agree or Agree*	172 (53.6)
*Strongly disagree or Disagree*	148 (46.3)
**It is preferable to acquire immunity against infectious diseases naturally (by having the disease) rather than by vaccination.**	
*Strongly agree or Agree*	90 (26.9)
*Strongly disagree or Disagree*	245 (73.1)
**The safety of a vaccine developed in an emergency, during an epidemic, cannot be considered guaranteed.**	
*Strongly agree or Agree*	123 (42.0)
*Strongly disagree or Disagree*	170 (58.0)
**I trust science to develop safe new vaccines.**	
*Strongly agree or Agree*	329 (96.8)
*Strongly disagree or Disagree*	11 (3.2)
**I trust science to develop effective new vaccines.**	
*Strongly agree or Agree*	326 (97.6)
*Disagree*	8 (2.4)
**Would you advise your patients to get vaccinated against COVID-19?**	
*Yes*	346 (98.0)
*No*	3 (0.9)
*Not sure*	4 (1.1)

## Data Availability

The datasets used and/or analyzed during the current study are available from the corresponding author on reasonable request.

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
