# Peer review of "Multi-Perspective Views and Hesitancy toward COVID-19 Vaccines: A Mixed Method Study"

_vaccines, 2023, doi:10.3390/vaccines11040801_

Round 1

Reviewer 1 Report

I will suggest to provide the following informations in the manuscript

1. Provide the details of the efforts taken by the government to increase the awareness and vaccine acceptance before the enrolment of the study participants. Inadequate public health efforts may increase the vaccine hesitancy

2.  Please add information about the access and use of electronic media and printed media by the 355 participants becuse this will be a major determinant for vaccine hesitancy

3. What was the prior exposure to Covid-19 in the participants and their close relative and what were their outcomes?

Author Response

On behalf of my co-authors, I am pleased to re-submit the revised manuscript entitled “Multi-perspective views and hesitancy towards COVID-19 vaccines: a mixed method study” for further consideration and inclusion in Vaccines. In addition to our point-by-point response to the 1st reviewer’s concerns which is provided below, we improved the description and clarity of the methods section and a native speaker has revised and improved the English language of the paper. All the changes are in tracks in the revised manuscript. Thank you for your time and constructive comments that helped us to improve the manuscript further.

I will suggest to provide the following informations in the manuscript

  1. Provide the details of the efforts taken by the government to increase the awareness and vaccine acceptance before the enrolment of the study participants. Inadequate public health efforts may increase the vaccine hesitancy

Thank you for the comment. We have briefly discussed the efforts taken by the government in terms of awareness raising campaigns in the introduction section.

 “The Ministry of Health (MoH) established a working group to coordinate a public awareness campaign when vaccines became available in Armenia. Comprised of over ten national and international organizations, it focused on developing risk communication tools, and social media campaigns to promote vaccination. The campaign messaging was delivered through various channels including ads on TV and social media.”

  1. Please add information about the access and use of electronic media and printed media by the 355 participants because this will be a major determinant for vaccine hesitancy

Thank you for the comment. We unfortunately did not collect data from the survey participants regarding their access to electronic and printed media but we improved the introduction section adding relevant information. Please see lines 88 and the same text below.

The government also initiated series of healthcare provider training focusing on the importance of vaccination, brand-specific aspects of vaccine administration, and possible side effects. The trainings particularly targeted physicians and nurses from the primary health care level.”

  1. What was the prior exposure to Covid-19 in the participants and their close relative and what were their outcomes?

Thank you for the comment. In terms of participants prior exposure to COVID-19 we had relevant data from both the survey and the qualitative interviews. We added the relevant information from the survey data under the results, theme 1 as follows (please see lines 290 and 270):

When asked about their prior exposure to COVID-19, 4.4% of the survey participants indicated having had the disease and being immune to it and 29.5% answered stated that they have already recovered from the disease and will not get re-infected.

As for the qualitative interviews, we have discussed participants’ preference of natural infection over the vaccines in our first theme. To highlight this preference a bit more we added these two quotes from the participants under the same theme: “1. We have already been infected, what’s the need for vaccination? 2. I know someone who was vaccinated and is currently hospitalized. It has been a month and I am not sure what will happen. That person also had COVID, took it very lightly, but now is in a very serious condition because of the vaccine.”

Reviewer 2 Report

General note: all abbreviation must be explanined and clear description should be added at the time of very first use of the given abbreviation

English proofreading is needed.

Specific Notes:

All quoted personal notes should be eliminnated as it is reducing the scientific soundness of the paper, sometimes refering just speculation such as "“I think if 60-70% [of the society] has immunity [having COVID-19 antibody as a result of contracting the disease]," --- no data, what survey is referred here, thus, it is sheer speculation, not worth referring in your paper.

Generally, 

Discussion: there is a sentence where exact reference is missing -

"....vaccines were widely available in the country (ref)." --- pls add reference as number

"A study conducted in Denmark revealed that the average vaccination rate among doctors with a positive attitude towards vaccination was 85%, compared to 69% in practices with more reserved attitudes [36]" - it must be highlighted it is not about Covid-19 vaccination, but about MMR vaccines.

"Physicians’ attitudes towards the COVID-19 vaccination can shape their readiness to properly guide patients in their decision making [41], and impact the public’s perception" - it is true but very obvious

In the discussion you should address the very interesting controversies such as :

A COVID-19 vaccination should be mandatory for everyone who is able to have it. Strongly agree or Agree 269 (76.6) Strongly disagree or Disagree 82 (23.4) 

VS

COVID-19 vaccines are being rushed without appropriate testing. Strongly agree or Agree 172 (53.6) Strongly disagree or Disagree 148 (46.3)

or

COVID-19 vaccines are being rushed without appropriate testing. Strongly agree or Agree 172 (53.6) Strongly disagree or Disagree 148 (46.3)

VS

I trust science to develop safe new vaccines. Strongly agree or Agree 329 (96.8) Strongly disagree or Disagree 11 (3.2)

or

Would you advise your patients to get vaccinated for COVID-19? Yes 346 (98.0) No 3 (0.9) Not sure 4 (1.1)

vs

COVID-19 vaccines are being rushed without appropriate testing. Strongly agree or Agree 172 (53.6) Strongly disagree or Disagree 148 (46.3)

COVID-19 vaccines are being rushed without appropriate testing. Strongly agree or Agree 172 (53.6) Strongly disagree or Disagree 148 (46.3)

vs

The safety of a vaccine developed in an emergency, during an epidemic, cannot be considered guaranteed. Strongly agree or Agree 123 (42.0) Strongly disagree or Disagree 170 (58.0)

Personally I am very much interested if it was any official (governmental or Chief Medical Officer) communication on Covid-19 cases and suggesting of Covid-19 vaccination for the public? 

It could be also an interesting from where physiscians obtained their information on Covid-19 which obviously influenced their attitude to Covid-19 vaccines. Was any official source of informaton?

Author Response

On behalf of my co-authors, I am pleased to re-submit the revised manuscript entitled “Multi-perspective views and hesitancy towards COVID-19 vaccines: a mixed method study” for further consideration and inclusion in Vaccines. In addition to our point-by-point response to the 1st reviewer’s concerns which is provided below, we improved the description and clarity of the methods section and a native speaker has revised and improved the English language of the paper. All the changes are in tracks in the revised manuscript. Thank you for your time and constructive comments that helped us to improve the manuscript further.

General note: all abbreviation must be explained and clear description should be added at the time of very first use of the given abbreviation

Thank you for your comment. We ensured that all abbreviations are explained upon their first use in the manuscript.

English proofreading is needed.

One of our co-authors who is a native English speaker proofread and improved the text throughout the manuscript. 

Specific Notes:

All quoted personal notes should be eliminated as it is reducing the scientific soundness of the paper, sometimes referring just speculation such as "“I think if 60-70% [of the society] has immunity [having COVID-19 antibody as a result of contracting the disease]," --- no data, what survey is referred here, thus, it is sheer speculation, not worth referring in your paper.

Thank you for pointing out this issue. Given that the main message of the quote is not necessarily fixating on the 60-70%, we used different wording in the following way to avoid speculation: “I think if the society [some adequate percentage of the society] has immunity [having COVID-19 antibody as a result of contracting the disease] ], vaccinations will not be as effective, they can even have a negative/reverse effect”.

Generally,

Discussion: there is a sentence where exact reference is missing -

"....vaccines were widely available in the country (ref)." --- pls add reference as number

            Thank you! We added the reference.

"A study conducted in Denmark revealed that the average vaccination rate among doctors with a positive attitude towards vaccination was 85%, compared to 69% in practices with more reserved attitudes [36]" - it must be highlighted it is not about Covid-19 vaccination, but about MMR vaccines.

We added the word MMR in the sentence to make it more clear: “A study conducted in Denmark revealed that the average Measles, Mumps, and Rubella (MMR) vaccination rate among doctors with a positive attitude towards vaccination was 85%, compared to 69% in practices with more reserved attitudes [39]

"Physicians’ attitudes towards the COVID-19 vaccination can shape their readiness to properly guide patients in their decision making [41], and impact the public’s perception" - it is true but very obvious

            Thank you. We deleted that particular sentence.

In the discussion you should address the very interesting controversies such as:

 A COVID-19 vaccination should be mandatory for everyone who is able to have it. Strongly agree or Agree 269 (76.6) Strongly disagree or Disagree 82 (23.4)

VS

COVID-19 vaccines are being rushed without appropriate testing. Strongly agree or Agree 172 (53.6) Strongly disagree or Disagree 148 (46.3)

 or

 COVID-19 vaccines are being rushed without appropriate testing. Strongly agree or Agree 172 (53.6) Strongly disagree or Disagree 148 (46.3)

VS

I trust science to develop safe new vaccines. Strongly agree or Agree 329 (96.8) Strongly disagree or Disagree 11 (3.2)

 or

 Would you advise your patients to get vaccinated for COVID-19? Yes 346 (98.0) No 3 (0.9) Not sure 4 (1.1)

vs

COVID-19 vaccines are being rushed without appropriate testing. Strongly agree or Agree 172 (53.6) Strongly disagree or Disagree 148 (46.3)

 COVID-19 vaccines are being rushed without appropriate testing. Strongly agree or Agree 172 (53.6) Strongly disagree or Disagree 148 (46.3)

vs

The safety of a vaccine developed in an emergency, during an epidemic, cannot be considered guaranteed. Strongly agree or Agree 123 (42.0) Strongly disagree or Disagree 170 (58.0)

Thank you for a great comment! We extended the text under the discussion section and highlighted the mentioned controversies. Please see lines 428, and the same text below:

“The majority of surveyed PHC providers mentioned trusting science to develop vaccines and advising their patients to get the COVID-19 vaccines. Yet, surprisingly nearly half of them agreed with the contradictory statement that the development of COVID-19 vaccines was rushed without appropriate testing. The literature shows varying patterns of perception regarding COVID-19 vaccination among healthcare providers, ranging from full acceptance to total refusal [40-43]. The inconsistencies and controversies that arose in our study in the attitudes towards COVID-19 vaccines are likely multifactorial. For example, the background, education, and professional practice of healthcare providers likely predisposed them to trust science and to consider vaccines an effective measure against infectious diseases. However, knowledge gaps regarding COVID-19 vaccine development on the one hand, and consistent and targeted misinformation from the media against COVID-19 vaccines on the other, might have triggered doubts and hesitancy. The idea that the COVID-19 vaccines had not yet undergone proper testing was amongst the most frequently cited arguments against COVID-19. In our findings, the influence of misinformation was also recognized by the participants.”

Personally I am very much interested if it was any official (governmental or Chief Medical Officer) communication on Covid-19 cases and suggesting of Covid-19 vaccination for the public?

Thank you for your interest. Right after the pandemic was declared, a single informational portal (Unified Infocenter) was established to communicate daily updates on the COVID-19 identified cases, hospitalization and mortality daily. The statistics were broadcasted on TV as well as published through official social media. To promote vaccination among the population the government organized an awareness-raising campaign and we mentioned that in the text. Perhaps that was not enough, or was a bit late.

It could be also an interesting from where physicians obtained their information on Covid-19 which obviously influenced their attitude to Covid-19 vaccines. Was any official source of information?

The physicians underwent a series of trainings on COVID-19 right from the beginning of the pandemic. The trainings were organized by the National Institutes of Health, Ministry of Health. Those official sources were provided to them. Apart from this they could be under the influence of misinformation that it is not possible to control. But we did not collect any specific data during the survey to look to include in the paper.

Round 2

Reviewer 1 Report

Thnaks for your response